# A Conditional Randomization Test for Sparse Logistic Regression in High-Dimension

**Binh T. Nguyen**
LTCI, Télecom Paris, IP Paris
tuanbinhs@gmail.com

**Bertrand Thirion**
Université Paris-Saclay, Inria, CEA, Palaiseau 91120, France
bertrand.thirion@inria.fr

**Sylvain Arlot**
Université Paris-Saclay, CNRS, Inria, Laboratoire de mathématiques d'Orsay, 91405, Orsay, France
sylvain.arlot@universite-paris-saclay.fr

## Abstract

Identifying the relevant variables for a classification model with correct confidence levels is a central but difficult task in high-dimension. Despite the core role of sparse logistic regression in statistics and machine learning, it still lacks a good solution for accurate inference in the regime where the number of features $p$ is as large as or larger than the number of samples $n$. Here we tackle this problem by improving the Conditional Randomization Test (CRT). The original CRT algorithm shows promise as a way to output p-values while making few assumptions on the distribution of the test statistics. As it comes with a prohibitive computational cost even in mildly high-dimensional problems, faster solutions based on distillation have been proposed. Yet, they rely on unrealistic hypotheses and result in low-power solutions. To improve this, we propose *CRT-logit*, an algorithm that combines a variable-distillation step and a decorrelation step that takes into account the geometry of the $\ell_1$-penalized logistic regression problem. We provide a theoretical analysis of this procedure, and demonstrate its effectiveness on simulations, along with experiments on large-scale brain-imaging and genomics datasets.

## 1 Introduction

Logistic regression is one of the most popular tools in modern applications of statistics and machine learning, partly due to its relative algorithmic simplicity. The method belongs to the class of *generalized linear models* that handle discrete outcomes, *i.e.* classification problems. Here, we focus on the binary classification problem, where one observation of the responses $y \in \{0, 1\}$ and the data vectors $\mathbf{x} \in \mathbb{R}^p$ follows the relationship:

$$\mathbb{P}(y = 1 \mid \mathbf{x}) = g(\mathbf{x}^T \boldsymbol{\beta}^0) = \frac{1}{1 + \exp(-\mathbf{x}^T \boldsymbol{\beta}^0)}, \tag{1}$$

where $g(x) = 1/(1 + \exp(-x))$ is the sigmoid function, and $\boldsymbol{\beta}^0$ the vector of true regression coefficients. In the classical setting, in which the number of samples $n$ is greater than the number of features $p$, an estimate $\hat{\boldsymbol{\beta}}$ of the true signals $\boldsymbol{\beta}^0$ can be obtained using *maximum likelihood estimation* (MLE). The asymptotic behaviour and derivation of the test statistic, confidence intervals and p-values of the MLE have been well studied, *e.g.* in [13]. The availability of p-values for the test statistics makes it possible to rely on multiple hypothesis testing, where one wants to test which variables have a non-zero effect on the outcome, *conditioned* on the remaining variables. Unfortunately, this line of analysis cannot be applied to the high-dimensional regime, where $p$ is larger than $n$, as argued in [25, 32, 34]. These works show that in the regime $\lim_{n,p\to\infty} n/p = \kappa$, the MLE estimator

exists only when $\kappa > 2$. However, we note that this type of analysis is done *without* the addition of $\ell_1$-regularization to the likelihood function, *i.e.* without using a *penalized estimator* to enforce sparsity.

**Motivation** Our focus in this paper is to do inference with statistical guarantees on high-dimensional sparse logistic regression, where $p$ is larger or much larger than $n$. This setting is typical in modern applications of pattern recognition, *e.g.* in brain-imaging or genomics [3], with $p$ as large as hundreds of thousands –compressible to thousands– but $n$ stays at most few thousand.

The family of methods we consider is the *Conditional Randomization Test (CRT)* [10]. CRT relies on generating multiple noisy copies of original variables to output empirical p-values in high-dimensional inference problems. However, prohibitive computational cost makes CRT impractical, as discussed at length in [10, 26, 7, 19]. There have been several lines of research attempting to fix this problem, most notably the *distilled Conditional Randomization Test (dCRT)* [19]. This work introduced a *distillation step* as a replacement for the randomized sampling step to compute the importance statistics (see Section 2 for more details). It provides a way to output p-values for multiple types of regression and classification problems, assuming convergence to Gaussian distribution of the test statistic in large-sample regime. Yet, as shown in the left panel of Figure 1, the originally proposed dCRT test-statistic for logistic regression does not behave as well as intended. *In particular, its null distribution deviates markedly from standard normal in high-dimension whenever $n/p \leq 1$.*

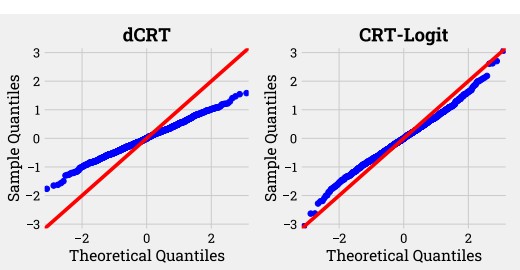

Figure 1: **QQ-Plot for 1000 samples of test-statistic of a null index for logistic regression**, with simulated data, $n = 200, p = 400$. *Left:* Statistics obtained from running Distilled-CRT, and *Right:* from our proposed algorithm. The empirical distribution of the dCRT null-statistic strays far from theoretical distribution, which is standard normal, while empirical distribution of CRT-logit's null test score is much closer.

**Contribution** We propose a correction for the dCRT, inspired by the decorrelation method presented in [21]. The decorrelation step makes the null-distribution of the test statistics much closer to standard normal, as shown on the right panel of Figure 1, and thus increases the statistical power of the method. We provide asymptotic analysis of this method, which shows that CRT-logit produces standard normal test-statistics in the large-sample regime. In addition, we validate the high performance of CRT-logit on large-scale brain-imaging and genetics datasets, thus showing its usefulness in practical applications.

**Related works** The closest cousin of the Conditional Randomization Test is Knockoff Filter [4, 10], a recent breakthrough in the False Discovery Rate (FDR) control literature. It relies on the creation of additional noisy features, called knockoffs, to calculate variable-importance statistics. Another extension of vanilla CRT is the Holdout Randomization Test (HRT) [26]. While still requiring multiple samplings of noisy variables, HRT solves the computational issue of original CRT by doing heavy model fitting only once on one part of the dataset, and test statistics calculation on the other part, without refitting the model. However, this method relies on sample-splitting, and hence inherently suffers from a loss of statistical power. A parallel line of work has introduced the Conditional Permutation Test (CPT) [7], a non-parametric alternative to CRT that relies on a random shuffling mechanism applied to original variables, instead of multiple sampling of new variables. This potentially makes CPT more robust to model mis-specification. [32] recently proposed a method called SLOE, which adapts the analysis of [34], but in a regime different from what we are considering, where $\lim_{n,p\to\infty} n/p \to \kappa \in (1,2)$, and more importantly without sparsity-inducing penalty. On a separate note, we notice the similarity of dCRT [19] with debiased Lasso [16, 28, 33]. This line of work proposed a debiasing formula for the estimator, which makes the asymptotic distribution of $(\hat{\beta}^{\text{LASSO}} - \beta^0)$ standard normal, so that one can compute the test statistic and p-value associated with each variable.

## 2 Background

**Notation** We denote matrices, vectors, scalars and sets by bold uppercase, bold lowercase, script lowercase , and calligraphic letters, respectively, *e.g.* $\mathbf{X}$, $\mathbf{x}$, $x$, and $\mathcal{X}$. The $i$-th row of a matrix $\mathbf{X}$ will be denoted $\mathbf{X}_{i,*}$ , the $j$-th column $\mathbf{X}_{*,j}$ and the $(i, j)$-th element $\mathbf{X}_{i,j}$. For any natural number $p$, we denote the set $[p] \overset{\text{def.}}{=} \{1, \ldots, p\}$. For each $\mathbf{x} \in \mathbb{R}^p$ and $j \in [p]$, we denote $\mathbf{x}_{-j} \overset{\text{def.}}{=} \{x_1, x_2, \ldots, x_{j-1}, x_{j+1}, \ldots, x_p\}$ a $p-1$ dimensional observation after removing the $j$-th variable. Correspondingly, $\mathbf{X}_{-j}$ is the data matrix $\mathbf{X} \in \mathbb{R}^{n \times p}$ with column $\mathbf{X}_{*,j}$ removed. The cumulative distribution function (CDF) of the standard Gaussian distribution will be denoted $\Phi(\cdot)$. The indicator function of a random event $\mathcal{A}$ will be denoted $\mathbf{1}_{\mathcal{A}}$. For any two positive sequences $x_n$ and $y_n$, we write $x_n \asymp y_n$ if $c y_n \leq x_n \leq C y_n$ for all $n$, for some positive constants $c$ and $C$. For a vector $\mathbf{x}$, $\|\mathbf{x}\|_p$ denotes its $\ell_p$ norm. For a function $f : \mathbb{R}^p \to \mathbb{R}$, $\nabla_j f$ denotes its gradient *w.r.t.* the $j$-th variable, for $j \in [p]$.

**Problem setting** We consider exclusively binary classification, where the response vector is denoted $\mathbf{y} \in \{0, 1\}^n$ and the data matrix $\mathbf{X} \in \mathbb{R}^{n \times p}$ consists of $n$ $p$-dimensional samples. Throughout the paper, we assume the data $\{\mathbf{X}_{i,*}\}_{i=1}^n$ are *i.i.d.* and follow a distribution with zero mean and population covariance matrix $\boldsymbol{\Sigma}$. Moreover, we assume that $\mathbf{X}_{i,*}$ and $\mathbf{y}_i$ follow the logistic relationship in Eq. (1). We denote the support set $\mathcal{S} \overset{\text{def.}}{=} \{j \in [p] : \boldsymbol{\beta}_j^0 \neq 0\}$ and assume that it is sparse, *i.e.* $\text{card}(\mathcal{S}) = s^* \ll p$, where $\text{card}$ denotes the cardinality of a set. Furthermore, $\hat{\mathcal{S}} \overset{\text{def.}}{=} \{j \in [p] : \hat{\beta}_j \neq 0\}$ indicates an estimation of $\mathcal{S}$, where $\hat{\beta}_j$ is an estimate of the true signal $\boldsymbol{\beta}_j^0$. We try to obtain it through a $\ell_1$-penalized logistic estimator:

$$\hat{\boldsymbol{\beta}}_\lambda = \underset{\boldsymbol{\beta} \in \mathbb{R}^p}{\text{argmin}} \ \ell(\boldsymbol{\beta}) + \lambda \|\boldsymbol{\beta}\|_1 \ , \ \text{with} \ \ell(\boldsymbol{\beta}) = -\frac{1}{n} \sum_{i=1}^n \left\{ (\mathbf{X}_{i,*}\boldsymbol{\beta})y_i - \log\left[1 + \exp(\mathbf{X}_{i,*}\boldsymbol{\beta})\right] \right\} \ . \quad (2)$$

We denote $\mathbf{I} \overset{\text{def.}}{=} \mathbb{E}_{\boldsymbol{\beta}^0}[\nabla^2 \ell(\boldsymbol{\beta}^0)]$ the Fisher information matrix, and $\mathbf{I}_{j|-j}$ the partial Fisher information, defined by $\mathbf{I}_{j|-j} \overset{\text{def.}}{=} \mathbb{E}[\nabla_{jj}^2 \ell(\boldsymbol{\beta}^0) - [\nabla_{j,-j}^2 \ell(\boldsymbol{\beta}^0)]^\top [\nabla_{-j,-j}^2 \ell(\boldsymbol{\beta}^0)]^{-1} \nabla_{-j,j}^2 \ell(\boldsymbol{\beta}^0)] = \mathbf{I}_{jj} - \mathbf{I}_{j,-j} \mathbf{I}_{-j,-j}^{-1} \mathbf{I}_{-j,j}$ , where $\mathbf{I}_{j,-j}$ is the row-vector made with the $j$th-row and the columns corresponding to $\boldsymbol{\beta}_{-j}$, $\mathbf{I}_{-j,-j}$ the sub-matrix of $\mathbf{I}$ made with the rows and columns corresponding to $\boldsymbol{\beta}_{-j}$. This quantity, defined following [13, pp. 323], plays an important role in our proposed method, detailed in Section 3.

**Statistical control with False Discovery Rate** To quantify statistical errors, we consider the *False Discovery Rate*, introduced in [5]. Given an estimate of the support $\hat{\mathcal{S}}$, the false discovery proportion (FDP) is the ratio of the number of selected features that do not belong to the true support $\mathcal{S}$, divided by the total number of selected features. The False Discovery Rate is the expectation of the FDP:

$$\text{FDP}(\hat{\mathcal{S}}) = \frac{\text{card}(\{j : j \in \hat{\mathcal{S}}, j \notin S\})}{\text{card}(\hat{\mathcal{S}}) \vee 1} \qquad \text{and} \qquad \text{FDR}(\hat{\mathcal{S}}) = \mathbb{E}[\text{FDP}(\hat{\mathcal{S}})].$$

**Conditional Randomization Test (CRT) and Distillation CRT (dCRT)** The concept of Conditional Randomization Test was originally proposed in the model-X knockoff paper [10] as a way to output valid empirical p-values using knockoff variables. The principle of the knockoff filter is first to sample noisy copies $\tilde{\mathbf{X}}_{*,j}$ of variable $\mathbf{X}_{*,j}$, given a known sampling mechanism $P_{j \ |-j}$. One advantage of the knockoff filter is that no specific assumption is placed on the distribution of the inferred test statistic. However, this means that, in general, there is no mechanism to derive p-values from the knockoff statistic. This motivates the introduction of CRT, which requires running high-dimensional inference for each variable $j$ $B$ times. However, the computation cost of CRT is prohibitive when $p$ grows large: assuming that we use the Lasso program with coordinate descent to compute $T_j^{\text{CRT}}$, its runtime would be $\mathcal{O}(Bp^4)$ [15, pp. 93]. Moreover, CRT requires decently large $B$ to make the empirical distribution of the p-values smooth enough. Reducing the computational cost of CRT is the main motivation of several works [7, 19, 26]. One of them is the introduction of distillation-CRT (dCRT) [19]. The main appeal of this method is that it can output p-values analytically, therefore bypassing the multiple knockoffs sampling steps used to infer on each variable, and leads to a reasonable reduction of the computation cost.

**Distillation operation** The key addition of dCRT is the distillation operation: for each variable $j$, we want to distill all the conditional information of the remaining variables $\mathbf{X}_{-j}$ to $\mathbf{x}_j$ and to $\mathbf{y}$ via least-squares minimization with $\ell_1$-regularization to enforce sparsity. For each variable $j$, we first solve the lasso problem by regressing $\mathbf{X}_{*,j}$ on $\mathbf{X}_{-j}$,

$$\hat{\boldsymbol{\beta}}^{d\mathbf{x}_{*,j}} = \underset{\boldsymbol{\beta} \in \mathbb{R}^{p-1}}{\operatorname{argmin}} \frac{1}{2} \|\mathbf{X}_{*,j} - \mathbf{X}_{-j}\boldsymbol{\beta}\|_2^2 + \lambda_{dx}\|\boldsymbol{\beta}\|_1. \tag{3}$$

For distillation of variable $j$ and the binary response $\mathbf{y}$ with logistic relationship, [19] briefly suggested to solve a penalized estimation problem, similar to Eq. (2):

$$\hat{\boldsymbol{\beta}}^{d_y,j} = \underset{\boldsymbol{\beta} \in \mathbb{R}^{p-1}}{\operatorname{argmin}} -\frac{1}{n} \sum_{i=1}^{n} \left\{ (\mathbf{X}_{i,-j}^\top \boldsymbol{\beta}) y_i - \log\left[1 + \exp(\mathbf{X}_{i,-j}^T \boldsymbol{\beta})\right] \right\} + \lambda\|\boldsymbol{\beta}\|_1. \tag{4}$$

Then, a test statistic is calculated for each $j = 1, \ldots, p$:

$$T_j = \sqrt{n} \, \frac{\langle \mathbf{y} - \mathbf{X}_{-j}\hat{\boldsymbol{\beta}}^{d_y,j}, \mathbf{X}_{*,j} - \mathbf{X}_{-j}\hat{\boldsymbol{\beta}}^{d\mathbf{x}_{*,j}} \rangle}{\|\mathbf{y} - \mathbf{X}_{-j}\hat{\boldsymbol{\beta}}^{d_y,j}\|_2 \|\mathbf{x}_j - \mathbf{X}_{-j}\hat{\boldsymbol{\beta}}^{d\mathbf{x}_{*,j}}\|_2}. \tag{5}$$

Intuitively, Eq. (5) is the correlation of the regression residuals, calculated from Eq. (3) and (4), scaled by a factor of $\sqrt{n}$. Under the null hypothesis, and more importantly, assuming linear relationship between $\mathbf{X}_{i,*}$ and $\mathbf{y}$, this quantity follows standard normal distribution asymptotically, conditional to $\mathbf{y}$ and $\mathbf{X}_{-j}$. It then follows that we can output a p-value for each variable $j$ by taking $\hat{p}_j = 2\left[1 - \Phi\left(T_j\right)\right]$.

However, the formulation of test statistics in Eq. (5) is not truly satisfactory in the setting of sparse logistic regression. More specifically, both the calculation of regression residuals $\mathbf{y} - \mathbf{X}_{-j}\hat{\boldsymbol{\beta}}^{d_y,j}$ and test statistics $T_j$ *do not take into account the non-linear relationship* between $\mathbf{X}$ and the binary response $\mathbf{y}$. The first row of Figure 6 plots the qq-plot of the test statistics $T_j$ for logistic regression, which shows that even in the classical regime where $n > p$, its distribution is far from standard normal.

## 3 Decorrelating Test-Statistics for High-Dimensional Logistic Regression

As we have elaborated, the formulation of dCRT is not well-suited for problems other than penalized least-squares regression. We therefore propose an adaptation of dCRT in the case of logistic regression, inspired by the classical work of [13] and by [21]. First, note that when testing $H_0^j : \beta_j^0 = 0$ under the case where $n > p$, we have the classical Rao's test statistic, defined by

$$T_j^{\text{Rao}} = \sqrt{n} \, \hat{\mathbf{I}}_{j|-j}^{-1/2} \nabla_j \ell(0, \hat{\boldsymbol{\beta}}_{-j}), \tag{6}$$

where $\nabla_j \ell(0, \hat{\boldsymbol{\beta}}_{-j}) \overset{\text{def.}}{=} \nabla_{\beta_j} \ell(\beta_j, \hat{\boldsymbol{\beta}}_{-j})\big|_{\beta_j=0}$ is the Fisher score. Here $\hat{\boldsymbol{\beta}}_{-j} \overset{\text{def.}}{=} \operatorname{argmin}_{\boldsymbol{\beta}_{-j} \in \mathbb{R}^{p-1}} \ell(\beta_j, \boldsymbol{\beta}_{-j})$ is the constrained maximum-likelihood estimator of $\boldsymbol{\beta}_{-j}$ with fixed $\beta_j$, and $\hat{\mathbf{I}}_{j|-j}$ is a consistent estimator of the partial Fisher information $\mathbf{I}_{j|-j}$. The appearance of the term $\hat{\mathbf{I}}_{j|-j}^{-1/2}$ is due to the fact that under the null hypothesis $H_0^j$, we have, by [13, Chapter 9], [24],

$$\sqrt{n}\nabla_j \ell(0, \hat{\boldsymbol{\beta}}_{-j}) \xrightarrow[n\to\infty]{(d)} \mathcal{N}(0, \mathbf{I}_{j|-j}),$$

which makes the asymptotic distribution of $T_j^{\text{Rao}}$ standard normal. However, in the high-dimension case, where $n < p$, we do not reach this convergence in distribution. To see this, consider the Taylor expansion of the Fisher score of variable $j$ around any given estimator $\widetilde{\boldsymbol{\beta}}_{-j}$ of the true $\boldsymbol{\beta}_{-j}^0$:

$$\nabla_j \ell(0, \widetilde{\boldsymbol{\beta}}_{-j}) = \nabla_j \ell(0, \boldsymbol{\beta}_{-j}^0) + \nabla_{j,-j}^2 \ell(0, \boldsymbol{\beta}_{-j}^0)(\widetilde{\boldsymbol{\beta}}_{-j} - \boldsymbol{\beta}_{-j}^0) + \mathcal{O}\left((\widetilde{\boldsymbol{\beta}}_{-j} - \boldsymbol{\beta}_{-j}^0)^2\right) \tag{7}$$

On the right-hand side, the first term converges weakly to a normal distribution due to the Central Limit Theorem, the remainder term becomes negligible using $\ell_1$ regularization to induce sparsity, but the second term does not, due to estimation bias and sparsity effect of $\widetilde{\boldsymbol{\beta}}_{-j}$ [14].

**Adapting distillation operation for sparse logistic regression** Fortunately, Eq. (7) suggests that for each variable $j$, we can *debias* the Fisher score by correcting the impact of other terms. In particular, for each variable $j$, we replace the Fisher score by

$$\nabla_j \ell(0, \boldsymbol{\beta}_{-j}) - \mathbf{I}_{j,-j}\mathbf{I}_{-j,-j}^{-1}\nabla_{-j}\ell(0, \boldsymbol{\beta}_{-j}). \tag{8}$$

The inversion of the large matrix $\mathbf{I}_{-j,-j} \in \mathbb{R}^{(p-1)\times(p-1)}$ is computationally prohibitive, but we can estimate the term $\mathbf{I}_{j,-j}\mathbf{I}_{-j,-j}^{-1}$ straightforwardly by solving

$$\hat{\mathbf{w}}^j = \underset{\mathbf{w}\in\mathbb{R}^{p-1}}{\mathrm{argmin}} \frac{1}{2n}\sum_{i=1}^{n}\left[\nabla^2_{j,-j}\ell_i(\hat{\boldsymbol{\beta}}) - \mathbf{w}^T\nabla^2_{-j,-j}\ell_i(\hat{\boldsymbol{\beta}})\right]^2 + \lambda\|\mathbf{w}\|_1, \tag{9}$$

for each variable $j$, where $\hat{\boldsymbol{\beta}}$ is given with Eq. (2). Moreover, since we have the closed-form of the derivatives of the logistic loss $\ell(\hat{\boldsymbol{\beta}})$, a simple derivation from Eq. (9) suggests the following $x_j$-distillation operation, *adapted for logistic regression*:

$$\hat{\boldsymbol{\beta}}^{d\mathbf{x}_{*,j}} = \underset{\boldsymbol{\beta}\in\mathbb{R}^{p-1}}{\mathrm{argmin}} \frac{1}{n}\sum_{i=1}^{n} g''(\mathbf{X}_{i,*}\hat{\boldsymbol{\beta}})(\mathbf{X}_{i,j} - \mathbf{X}_{i,-j}\boldsymbol{\beta})^2 + \lambda_{dx}\|\boldsymbol{\beta}\|_1, \tag{10}$$

where the extra term (second-order derivative of the sigmoid function) $g''(\mathbf{X}_{i,*}\hat{\boldsymbol{\beta}}) = \frac{\exp(\mathbf{X}_{i,*}\hat{\boldsymbol{\beta}})}{[1+\exp(\mathbf{X}_{i,*}\hat{\boldsymbol{\beta}})]^2}$ appears from differentiating twice the loss function $\ell(\hat{\boldsymbol{\beta}})$, and $\hat{\boldsymbol{\beta}} = \hat{\boldsymbol{\beta}}_\lambda$ is defined in Eq. (2). On the other hand, we can obtain $\hat{\boldsymbol{\beta}}_j^{d_y,j}$ from $\hat{\boldsymbol{\beta}}$ by simply omitting the $j$-th coefficient from it, *i.e.*

$$\hat{\boldsymbol{\beta}}^{d_y,j} \overset{\text{def.}}{=} (\hat{\beta}_1, \hat{\beta}_2, \ldots, \hat{\beta}_{j-1}, \hat{\beta}_{j+1}, \ldots, \hat{\beta}_p).$$

Finally, the equation for decorrelated test score, adapted from both Eq. (5) and (6), reads

$$T_j^{\text{decorr}} = -\frac{1}{\sqrt{n}}\,\hat{\mathbf{I}}_{j|-j}^{-1/2}\,\sum_{i=1}^{n}\left[y_i - g(\mathbf{X}_{i,-j}\hat{\boldsymbol{\beta}}^{d_y,j})\right]\left[\mathbf{X}_{i,j} - \mathbf{X}_{i,-j}\hat{\boldsymbol{\beta}}^{d\mathbf{x}_{*,j}}\right], \tag{11}$$

where the formula for the empirical partial Fisher information is $\hat{\mathbf{I}}_{j|-j} = n^{-1}\sum_{i=1}^{n} g''(\mathbf{X}_{i,*}\hat{\boldsymbol{\beta}})(\mathbf{X}_{i,j} - \mathbf{X}_{i,-j}\,\hat{\boldsymbol{\beta}}^{d\mathbf{x}_{*,j}})\,\mathbf{X}_{i,j}$. A summary of the full procedure, which we call CRT-logit, can be found in Algorithm 1. Notice that the runtime of CRT-logit is the same as dCRT, which means in general slower than KO and HRT. To speedup inference time, we introduce a variable-screening step that eliminates potentially unimportant variables before distillation, similar to dCRT. We provide empirical benchmark of the runtime of each method in Section 4.5.

**Setting $\ell_1$-regularization parameter $\lambda$ and $\lambda_{dx}$** In general, we advise to use cross-validation for obtaining $\hat{\boldsymbol{\beta}}_\lambda$ in Eq. (2) and for $\mathbf{X}_{*,j}$-distillation operator, as defined by Eq. (10). This is in line with the theoretical argument for dCRT [19, Lemma 1 and Theorem 3]. However, we also observe empirically that choosing the $\ell_1$-regularization parameters of the distillation step can strongly affect how variables are selected by CRT-logit. We provide more details in the Supplementary Material, and leave further theoretical investigations of this phenomenon for future work.

---

**Algorithm 1:** CRT-logit

---

1   **INPUT** design matrix $\mathbf{X} \in \mathbb{R}^{n\times p}$, reponses $\mathbf{y} \in \mathbb{R}^n$
2   **OUTPUT** vector of p-values $\{p_j\}_{j=1}^p$;
3   $\hat{\boldsymbol{\beta}}_\lambda \leftarrow$ `solve_sparse_logistic_cv`$(\mathbf{X}, \mathbf{y})$ // Using Eq. (2)
4   $\hat{\mathcal{S}}^{\text{SCREENING}} \leftarrow \{j : j \in [p], \hat{\beta}_j \neq 0\}$
5   **for** $j \in \hat{\mathcal{S}}^{\text{SCREENING}}$ **do**
6     $\hat{\boldsymbol{\beta}}^{d\mathbf{x}_{*,j}} \leftarrow$ `solve_scaled_lasso_cv`$(\mathbf{X}_{*,j}, \mathbf{X}_{*,-j})$ // Using Eq. (10)
7     $\hat{\boldsymbol{\beta}}^{d_y,j} \leftarrow (\hat{\beta}_1, \hat{\beta}_2, \ldots, \hat{\beta}_{j-1}, \hat{\beta}_{j+1}, \ldots, \hat{\beta}_p)$
8     $T_j^{\text{decorr}} \leftarrow$ `decorrelated_test_score`$(j, \mathbf{X}, \mathbf{y}, \hat{\boldsymbol{\beta}}^{d\mathbf{x}_{*,j}}, \hat{\boldsymbol{\beta}}^{d_y,j})$ // Using Eq (11)
9     $\hat{p}_j \leftarrow 2[1 - |\Phi(T_j^{\text{decorr}})|]$
10   **end**
11   **for** $j \notin \hat{\mathcal{S}}^{\text{SCREENING}}$ **do**
12     $\hat{p}_j = 1$
13   **end**

---

**Asymptotic analysis of the Decorrelated Test Statistic** We now provide theoretical analysis of CRT-logit in large-sample regime. All the proofs can be found in the Supplementary Material. First, we introduce the following assumption.

**Assumption 3.1** (Regularity conditions). *Assume that*

(A1) $\lambda_{\min}(\mathbf{I}) \geq K$ *for some constant* $K > 0$.

(A2) *Sparsity of* $\boldsymbol{\beta}^0$ *and* $\mathbf{w}^{0,j}$*, with* $\mathbf{w}^{0,j}$ *the ground truth weights for the distillation of* $\mathbf{x}_j$ *in Eq. (10):* $|\mathcal{S}| = s^*$ *and* $\|\mathbf{w}^{0,j}\|_0 = s'$ *with* $s^* = o\big(n^{1/2}/\log(p)\big)$ *and* $s' = o\big(n^{1/2}/\log(p)\big)$.

(A3) *For all* $i \in [n]$, $\mathbf{X}_{i,*}$ *and* $(-y_i + g'(\mathbf{X}_{i,*}\beta))$ *are sub-exponential random variables, and* $|\mathbf{X}_{i,-j}\mathbf{w}^{0,j}| \leq K'$ *almost surely, for some constant* $K' > 0$.

We then have the following result, that states that the asymptotic distribution of the decorrelated test scores is standard normal.

**Theorem 3.1.** *Let* $j \in [p]$*, and let* $T_j^{decorr}$ *be defined as in Eq. (11), with* $\lambda \asymp \lambda_{dx} \asymp \sqrt{n^{-1}\log(p)}$. *Then, if Assumption 3.1 holds true, and if we consider* $p = p(n)$*, under the null hypothesis* $\mathcal{H}_0^j : \beta_j^0 = 0$*, we have*

$$\forall t \in \mathbb{R}, \qquad \lim_{n\to\infty} |\mathbb{P}_{\beta^0}(T_j^{decorr} \leq t) - \Phi(t)| = 0\,,$$

*where* $\Phi(\cdot)$ *is the CDF of the standard Gaussian distribution. Moreover, for each* $j \in [p]$*, if we define* $\widehat{p}_j \stackrel{def.}{=} 2\left[1 - \Phi(T_j)\right]$*, i.e.* $\widehat{p}_j$ *is the output of Algorithm 1, then, under the null hypothesis* $\mathcal{H}_0^j : \beta_j^0 = 0$*, we have*

$$\limsup_{n\to\infty} \mathbb{P}_{\beta^0}(\widehat{p}_j \leq t) \leq t \quad \text{for all } t \in [0,1]\,,$$

*that is, the p-values output by Algorithm 1 are valid asymptotically.*

**Remark 3.1.** *We also show in the proof of Theorem 3.1 (in Supplementary Material) that the rate of convergence is* $\mathcal{O}(1/\sqrt{n})$*. Compared with some of the related works (e.g., knockoffs) that come with finite sample guarantees, our theoretical analysis only works in the asymprotic regime. We leave the finite sample analysis as one of the directions for future works.*

**FDR control with CRT-logit** As a consequence of Theorem 3.1, we have the following result, which establishes that the FDR of the test is controlled when using the Benjamini-Yekutieli procedure [6] with the p-values output from Algorithm 1, assuming that the number of tests $p$ is fixed.

**Corollary 3.1.** *Under Assumptions 3.1 and logistic model defined in Eq (1), with* $\lambda \asymp \lambda_{dx} \asymp \sqrt{n^{-1}\log(p)}$*, assume* $n^{-1/2}(s' \vee s^*)\log(p) = o(1)$*, and assume the number of tests* $p$ *is fixed. Let* $\alpha \in (0,1)$ *and* $\widehat{\mathcal{S}}_{BY\text{-}CRT}$ *be given by applying following the Benjamini-Yekutieli FDR-controlling procedure to the CRT-logit p-values* $\{\widehat{p}_j\}_{j \in [p]}$*, output from Algo.1. Then, we have*

$$\limsup_{n\to\infty} \mathbb{E}\left[\frac{\mathrm{card}(\widehat{\mathcal{S}}_{BY\text{-}CRT} \cap \mathcal{S}^c)}{\mathrm{card}(\widehat{\mathcal{S}}_{BY\text{-}CRT}) \vee 1}\right] \leq \alpha\,.$$

**Remark 3.2.** *Assumption 3.1 is also assumed in [21, 28], which also provide a detailed discussion of this regularity assumption in generalized linear models. This assumption, in turn, is built on the regularity assumption in the classic work [13, Chapter 9] to establish asymptotic normality of Rao's test statistic. Theorem 3.1 is an adaptation of [21, Theorem 3.1], specialized for the case of sparse logistic regression and the p-values output from CRT-logit.*

## 4 Empirical Results

We provide benchmarks of the proposed CRT-logit algorithm along with most other methods mentioned in the introduction, in particular: model-X Knockoff (KO) [10], Debiased Lasso (dLasso) [33, 16], original CRT with 1000 samplings [10], Holdout Randomization Test with 5000 samplings [26], and Lasso-Distillation CRT (dCRT) [19]. We did not include SLOE and CPT as the provided open-source implementation are particularly unstable and do not fit in the sparse-regression setting (for SLOE), or implementation is not available (for CPT). For the lack of space, we leave the extra experiment with a genome-wide association study in the Supplementary Material.

**Remark 4.1.** *As a slight caveat, in the simulated and semi-realistic experiment sections (Sections 4.1, 4.2 and 4.3), we introduce an additional noise term to the logistic relationship of Eq. (1):*

$$\mathbb{P}(y_i = 1 \mid \mathbf{x}_i) = g(\mathbf{x}_i^T \boldsymbol{\beta}^0 + \sigma \xi_i)\,, \tag{12}$$

where $\xi_i \sim \mathcal{N}(0, 1)$ is a Gaussian noise and $\sigma > 0$ the noise magnitude. The formula in Eq. (12) has been used in previous works, e.g. [8]. There is a clear justification to this: in most of the applications we consider, data are collected with measurement errors. In the case of brain-imaging, for example, recording the brain signal of the human subjects by scanners often includes noise caused either from the machine, or from the movement of the subjects [18]. Moreover, in general, this setting corresponds to a model mis-specification, which the CRT-logit is robust to under Assumption 3.1, following the same argument as in [21, Section 5].

**Remark 4.2.** We use Benjamini-Hochberg step-up procedure [5] to control FDR with the p-values in all the empirical experiments in Section 4.2 and App. 4.4, as we observe that the FDR is empirically controlled with this procedure, without compromising power with the conservative BY bound.

## 4.1 Effectiveness of the decorrelation step

To show how decorrelating the test statistics helps, we set up a simulation with matrix $\mathbf{X}$ of $p = 400$ features and vary the number of samples $n \in \{200, 400, 800, 4000\}$. The binary response vector $\mathbf{y}$ is created following Eq. (12), and the design matrix $\mathbf{X}$ is sampled from a multivariate normal distribution with zero mean, while the covariance matrix $\mathbf{\Sigma} \in \mathbb{R}^{p \times p}$ is a symmetric Toeplitz matrix, where the parameter $\rho \in (0, 1)$ controls the correlation between neighboring features: correlation decreases quickly when the distance between feature indices increases. The true signal $\boldsymbol{\beta}^0$ is picked with a sparsity parameter $\kappa = s^*/p$ that controls the proportion of non-zero elements with magnitude 2.0, *i.e.* $\beta_j = 2.0$ for all $j \in \mathcal{S}$. For the specific purpose of this experiment, non-zero indices of $\mathcal{S}$ are kept fixed. The noise $\boldsymbol{\xi}$ is *i.i.d.* normal $\mathcal{N}(0, \mathbf{Id}_n)$ with magnitude $\sigma = \|\mathbf{X}\boldsymbol{\beta}^0\|_2/(\sqrt{n} \text{ SNR})$, controlled by the SNR parameter. In short, the three main parameters controlling this simulation are correlation $\rho$, sparsity degree $\kappa$ and signal-to-noise ratio SNR. We generate randomly 1000 datasets, and run dCRT and CRT-logit algorithm to obtain one sample of test statistics $\{T_j\}_{j=1}^p$ and $\{T_j^{\text{decorr}}\}_{j=1}^p$. Then, we pick 1000 samples of one null test statistic $T_j$ and $T_j^{\text{decorr}}$, defined in Eq. (5) and (11), respectively, and plot the qq-plot of their empirical quantile versus the standard normal quantile. From the results in Figure 6, we observe that the empirical null distribution of the test

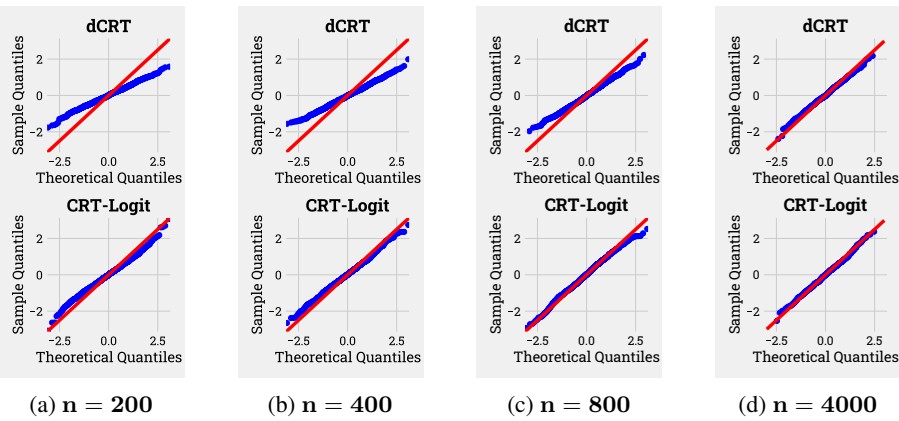

| (a) $\mathbf{n = 200}$ | (b) $\mathbf{n = 400}$ | (c) $\mathbf{n = 800}$ | (d) $\mathbf{n = 4000}$ |

Figure 2: **QQ-Plot for one null CRT statistic for logistic regression, with varying number of samples and a fixed number of variables** $p = 400$. The theoretical quantiles are obtained from a standard Gaussian distribution. The decorrelation step makes the empirical null distribution of the null statistics much closer to standard Gaussian. Parameters: SNR $= 3.0, \rho = 0.4$, sparsity $= 0.06$. *Upper row: Distilled-CRT statistic defined by Eq. (5). Bottom row: CRT-logit, with decorreleated test score defined by Eq. (11)* **(ours)**.

statistic is *much closer to a standard normal when adding the decorrelation step.* In particular, when the sample size $n$ increases to 400, the decorrelated test statistic has empirical quantiles almost inline with the theoretical quantiles of the standard normal distribution, while dCRT test score strays away from the 45-degree line. Again, we emphasize that the normality of $T_j$ is crucial for the p-values calculation. This outlines the importance of the decorrelating step on $T_j$.

## 4.2 High-dimensional scenario with varying simulation parameters

To see how each algorithm performs under different settings, we follow the same simulation scenario as in Sec. 4.1, but vary each of the three simulation parameters, while keeping the others unchanged

at a default value of SNR = 2.0, $\rho = 0.5$, $\kappa = 0.04$. We target a control of FDR at level 0.1, using the Benjamini-Hochberg procedure. The results in Figure 3 show that CRT-logit is the most powerful method while still controlling the FDR. Moreover, in the presence of higher correlations between nearby variables ($\rho > 0.6$), other methods suffer a drop in average power, but this is not as severe for CRT-logit. The original CRT, in general, is conservative. We believe that this is due to using only $B = 500$ samplings to generate empirical p-values for the two methods, due to prohibitive average runtime of the algorithm with larger $B$ (which we provide in Section 4.5). For HRT, the conservativeness is expected, due to the usage of only half of the sample for test-statistics calculation – even though the number of samplings is bigger than original CRT ($B = 5000$). We note that, perhaps surprisingly, the debiased lasso (`cdlasso`) is the most conservative. It controls FDR well in all settings. This might be due to the fact that `dlasso` also relies on the choice of the $\ell_1$-regularization $\lambda$ in the nodewise Lasso operation, similar to the $\mathbf{X}_{*,j}$-distillation of dCRT, as noted in Section 1. What makes the difference is that instead of using cross-validation for setting $\lambda$ for each variable $j$, a *fixed* value of $\lambda = 10^{-2}\lambda_{max}$ is used in the implementation of `dlasso`. We strongly suspect this fixed value is not optimal, which makes the procedure powerless.

## 4.3 Application: large-scale analysis on brain-imaging dataset

**Description**   The Human Connectome Project dataset (HCP) is a collection of brain imaging data on healthy young adult subjects with age ranging from 22 to 35. More specifically, the input $\mathbf{X}$ is a set of 2mm statistical maps of 400 subjects across 8 cognitive tasks. These are called z-maps, as the data follow a standard normal distribution under the null hypothesis. Each task in turn features 2 different contrasts, which effectively form binary responses $\mathbf{y} \in \{0,1\}^n$. In short, the goal of this fMRI data analysis is to identify voxels with task-related levels of activity by fitting $\mathbf{y}$ through distributed brain signals. The setting is high-dimensional with $n = 800$ samples, corresponding to 400 subjects, while the total number of variables is $p \approx 200,000$ brain voxels. Following [11, 20], we use a hierarchical clustering scheme to group the variables into $C = 1000$ spatially connected clusters. We provide details of the pre-processing step in Supplementary Material.

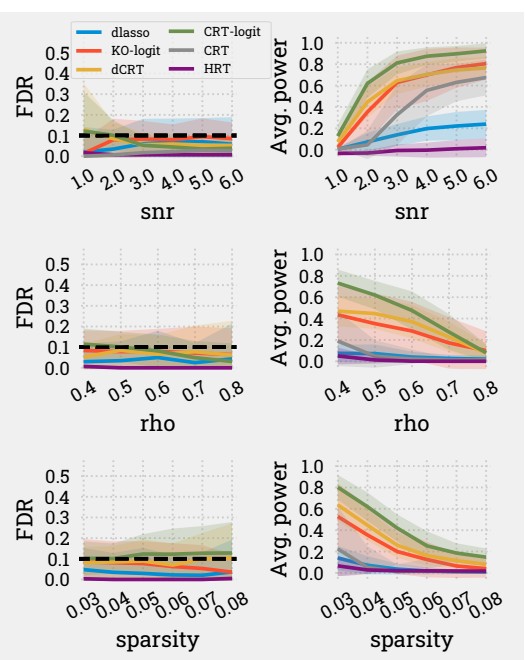

Figure 3: **FDR/Average Power of 100 runs of simulations across varying parameters in high-dimensional settings**.   Default parameter: $n = 400, p = 600, \text{SNR} = 2.0, \rho = 0.5, \kappa = 0.04$. FDR is controlled at level $\alpha = 0.1$. Methods: Debiased Lasso (`dlasso`), model-X Knockoff (`KO-logit`), original CRT (`CRT`), HRT (`HRT`), dCRT (`dCRT`), and our version of CRT (dark green line – `CRT-logit`).

**Creating semi-realistic ground-truth and response labels**   Since there is no ground truth for this dataset, we create synthetic true signals by fitting the data $\mathbf{X}$ and response $\mathbf{y}$ with an $\ell_1$-penalized logistic classifier. In other words, the estimator $\hat{\beta}^{\texttt{logreg}}$ will serve as true regression coefficients for each task. Then, to avoid bias in simulating label $\hat{\mathbf{y}}$, the z-maps matrix $\mathbf{X}$ of one task are used in conjunction with the discriminative pattern map $\hat{\beta}^{\texttt{logreg}}$ from the next task in the following order: `relational`, `gambling`, `emotion`, `social`. For instance, we use $\hat{\beta}^{\texttt{logreg}}$ of `gambling` with z-maps data matrix of `relational`, *i.e.* for all $i = 1, \ldots, n$, given $\mathbf{x}_{i,\texttt{relational}}$,

$$\hat{y}_i \sim \text{Bern}\left\{ g(\mathbf{x}_{i,\texttt{relational}}^\top \hat{\beta}_{\texttt{gambling}}^{\texttt{logreg}} + \sigma\xi_i) \right\}, \tag{13}$$

where $\text{Bern}(a)$ is a Bernoulli probability mass function that takes a value 1 with probability $a$, $\sigma$ is a noise magnitude and $\xi_i$ is a standard normal noise. Finally, we apply all inference algorithms on the semi-synthetic data $(\mathbf{X}, \hat{\mathbf{y}})$, and we evaluate their performance using the ground-truth $\hat{\beta}^{\texttt{logreg}}$. This simulation setting is similar to [11], except that here we consider a classification and not a regression

problem. It allows us to calculate the False Discovery Rate and average power with multiple runs of the inference procedure (across tasks).

**Remark 4.3.** *The* i.i.d. *assumption is formally violated in this experiment, where for each subject we analyze two sample images that are not independent. Yet, this remains a short-range correlation structure, and is thus not a strong challenge to the* i.i.d. *assumption.*

**Results** The results in Figure 4 show that CRT-logit achieves a better recovery compared to KO or original CRT/dCRT/HRT, which results in higher statistical power. This gain comes with a good control of the FDR under desired level $\alpha = 0.1$. On a related note, the only analysis where dCRT makes more discoveries than CRT-Logit is in the `emotion` task, but at the cost of failing to control FDR at the nominal level.

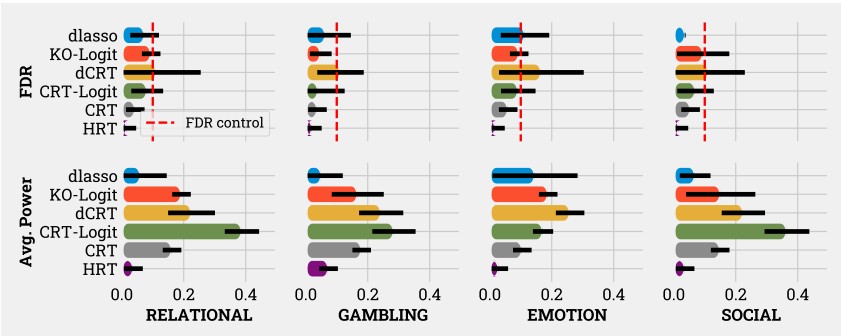

Figure 4: **FDR/Average Power of 50 runs of semi-realistic experiments on four tasks of Human Connectome Project dataset.** Parameters: $n = 800$ (taken from 400 subjects), SNR $= 2.0$. Methods (clustering versions): Debiased Lasso (`cdlasso`), model-X Knockoff (`cKO-logit`), original CRT (CRT), HRT (HRT), dCRT (dCRT), and our version of CRT (dark green line – `CRT-logit`).

## 4.4 Application: genome-wide association study with Human Brain Cancer Dataset

**Description** The last in our benchmark is a Genome-wide Association Study (GWAS) on the The Cancer Genome Atlas (TCGA) dataset [30, 31]. We choose to analyze the Glioma cohort, which consists of $n = 1026$ patients across a wide age range, diagnosed with this type of brain tumor, with a total of $p = 24776$ genes in the data matrix recorded as copy number variations (CNVs) at the gene level in log ratio format. As with the brain-imaging inference in Section 4.3, we use clustering to reduce the dimension to $C = 1000$ clusters. However, we use different criterion to merge variables (genes) to clusters of variables, which is the pairwise Linkage Disequilibrium, following [1, Section 4] (with available R library). For the response, a long-term survivor (LTS) is defined as a patient who survived more than five years after diagnosis and would be labeled $y = 0$, and otherwise would be a short-term survivor (STS), labeled $y = 1$. The objective is to identify significant genes that contribute to classification of the LTS/STS status. Similar to the Human Connectome Project dataset, there is no real ground-truth for the TCGA Glioma. However, we have the list of mutations and the frequency of those detected in the diagnosed patients. We therefore select the 1000 most frequent gene mutations that appeared in this list, *i.e.* the ground truth list consists of 1000 genes (variables).

Table 1: **List of detected genes associated with Glioma Cancer from the TCGA dataset.** $n = 1026$, $p = 24776$ (clustered to $C = 1000$). Empty line (—) signifies no detection. Methods listed in the table are the clustering version. Commonly detected genes between methods are put in bold text. Most detected genes are listed in the mutant list database that can be found in the recorded patients [30].

| Methods | Detected Genes |
| --- | --- |
| dLasso | — |
| KO | **ABCC10**, **ANK3**, CDH23, PTEN, **SPEN**, **SVIL**, ZMIZ1 |
| dCRT | **ANK3**, **ANKRD30A**, CDH23, PTEN, RET, **SPEN**,ZMIZ1 |
| CRT-logit | **ABCC10**, **ANKRD30A**, BCOR, EPHA3, PPL, SPAG17, **SPEN**, **SVIL**, USP9X |
| Original CRT | **ABCC10**, BCOR, EPHA3, **SPEN**, **SVIL** |
| HRT | **ABCC10**, **SPEN** |

**Result**  The result from Table 1 shows that CRT-logit finds the largest number of genes. Moreover, most of selected genes in this table are detected in the list of mutated genes found on recorded patients. Some genes are detected by all the benchmarked methods, most prominently SPEN, which is found on over 10 % of patients in the cohort. Furthermore, this gene is known to be associated not only with brain cancer, but also with other types of cancer in The Human Protein Atlas project [17]. Note that, in the absence of a ground-truth, this does not guarantee all genes found are associated with glioma, but this experiment demonstrates the capability of CRT-logit in GWAS studies.

## 4.5  Average runtime of benchmarked methods

Table 2: **Average runtime of benchmarked methods for one simulation (in seconds).** Standard error is reported in parentheses.

| Methods | Simulated (Sec. 4.2) | HCP-semi-real (Sec. 4.3) |
|---|---|---|
| Debiased Lasso [33, 28, 16] | 61.83 (5.2) | 154.27 (8.79) |
| Knockoff Filter [4, 10] | 1.62 (0.02) | 8.12 (0.62) |
| CRT (500 samplings) [10] | 2312.91 (38.21) | 7069.96 (109.09) |
| HRT (5000 samplings) [26] | 14.84 (2.01) | 52.17 (4.11) |
| dCRT[screening=True] [19] | 16.83 (1.89) | 65.18 (3.91) |
| dCRT[screening=False] [19] | 370.12 (8.18) | 962.40 (20.63) |
| **CRT-logit[screening=True] (this work)** | **14.16 (0.35)** | **61.26 (3.55)** |
| **CRT-logit[screening=False] (this work)** | **367.91 (4.11)** | **983.78 (17.26)** |

Besides statistical performance, it is equally important to assess the computational cost of inference procedures. The average runtime in Table 2 from the two experiments shows that the original CRT is not suitable for large-scale inference: it is over 2000 times slower than the fastest method (Knockoff Filter), and over 150 times slower than dCRT/CRT-logit. The empirical runtime also confirms the effectiveness of the screening step before doing distillation/decorrelation of the test-statistics: the step makes CRT-logit and dCRT 20 times faster than without. On a related note, although in theory Debiased Lasso, dCRT and CRT-logit (both without screening) share the same runtime complexity, the latter two are slower due to the use of cross-validation to estimate the sparsity hyperparameter $\lambda$ and $\lambda_{dx}$ (detailed in Section 3).

## 5  Discussion

We proposed an adaptation of the Conditional Randomization Test (CRT) for sparse logistic regression in the high-dimensional regime. A major improvement of CRT-logit, our proposed algorithm, compared to original CRT, comes from the decorrelation of test statistics to make their distribution closer to standard normal. Indeed, results from synthetic experiments in Figure 6 show that in high-dimension (when $0.5 \leq n/p \leq 1.0$), the empirical null distribution of CRT-logit's test statistic $T^{\mathrm{decorr}}$ is much more similar to a standard normal compared to the original CRT test statistic. Moreover, empirical benchmarks in Section 4 demonstrate that CRT-logit performs better than related statistical inference methods, such as the Debiased Lasso or Model-X Knockoffs. In particular, CRT-logit is the most powerful method in our synthetic experiment with high-dimensional datasets in Section 4.2, while still keeping FDR controlled under predefined level $\alpha = 0.1$ (with a slight caveat of using BH instead of BY procedure, as elaborated in Remark 4.2). We note that there exist some limitations to CRT-logit. The computational cost of CRT-logit, while lower than vanilla CRT, is still larger than alternative methods such as Knockoff Filter and Holdout Randomization Test. Moreover, tuning the $\ell_1-$regularization $\lambda_{dx}$ parameter by cross-validation, as is often done, can further increase the computational cost of CRT-logit (and dCRT). Despite this, our empirical benchmarks on both simulated and real data show real promises of CRT-logit. Henceforth, we believe CRT-logit is competitive for practical settings that involve structured data, such as brain-imaging and genomics applications.

**Acknowledgements**

BN, BT and SA acknowledged the support of the French "Agence Nationale de la Recherche" under the project ANR-17-CE23-0011 (FastBig) and ANR-20-CHIA-0025-01 (KARAIB AI chair). SA was also supported by Institut Universitaire de France (IUF), and BN was also supported by Research Chair DSAIDIS (Data Science and Artificial Intelligence for Digitalized Indutry and Services) of Télecom Paris.

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
