# OpenReview forum: "A Conditional Randomization Test for Sparse Logistic Regression in High-Dimension"
_NeurIPS.cc/2022/Conference — NeurIPS 2022 Accept_

### Official Review · Reviewer_np9G · 2022-06-22

**Rating:** 7
**Confidence:** 3
**Soundness:** 3 good
**Presentation:** 3 good
**Contribution:** 4 excellent

**Summary:**

This paper presents a method for performing hypothesis testing for lasso-penalized logistic regression in the high dimensional (but sparse) setting.  The main result relies on the asymptotic normality of a test statistic that is essentially how correlated a given feature is

**Questions:**

Comments / questions:

* Figure 2 is interesting, but the asymptotics presented here -- and the really interesting case -- is considering what happens as both n and p get large.  A very small simulation study to show that the asymptotic results hold even for p >> n would, in my opinion, strengthen the work.  Furthermore, in both the brain processing application and the TCGA application the data is preprocessed so that p ~ n.  Is this because the p^3 scaling makes it infeasible to run on the full dataset, or are there additional issues with p >> n?

* There is no discussion of finite sample guarantees.  e.g., Knockoffs control the FDR for finite sample sizes, whereas the present method only has such guarantees in the asymptotic regime.  This is not a big deal, but it would be good to mention this somewhere (maybe in the related works section).



Minor Comments / questions :
* The switch from $\hat{w}^j$ to $\hat{\beta}^{d_{x_{*,j}}$ from equations (9) to (10) is a bit notationally confusing.  Adding a sentence beforehand would help with the transition.

* In lines 171-174 the mention of a variable-screening step makes it sound like a somewhat heuristic procedure, when, in fact, it just leverages that it is trivial to compute p-values for j's where $\hat{beta}_j$ is zero.

* It would be nice to be notationally consistent between Assumption 3.1 and Assumption A.1. (e.g., $\kappa^2$ becomes $K$ and $K$ becomes $K'$)

* In Theorem 3.2, is p overloaded as both the number of features in the design matrix and also the number of tests? If separate they should not both be labeled $p$.  If they are the same, then isn't the theorem trivial by the assumption of $p$ being fixed (eventually $n \gg p$ and it is essentially just standard logistic regression)

* Remark 4.2 feels important and should be emphasized a bit more in my opinion -- the procedure used in the empirical tests is not proven to be valid by the analysis, but it seems to work in practice.

* In Table 1, the boldface should be removed.  The standard usage of boldface in these types of tables is to highlight the fastest method.  The parenthetical "(this work)" is already sufficient to draw the reader's attention to CRT-logit

* The notation in appendix A was a bit confusing to me.  In particular, in several places (e.g., the definition of $v^*$) a vector is defined which only makes sense later under the assumption that $j=1$.  For example $v^*$ is defined as ($1, -w^{0, j})$, but then ${v^*}^\top \nabla \ell(\beta)$ only makes sense if the 1 in $v^*$ lines up with $j$.  This is obviously not a big problem and doing something more "correct" might even be notationally clunky enough to not be worth it, but at the very least a quick explanation of the abuse of notation would be helpful.

* $\hat{v}$ has not yet been defined by Lemma A.4.

* $v^*$ as defined on line 404 and defined (again) on line 414 have different signs for the $w^{0, j}$ part.

* I may be missing something, but is some term missing from the first line in the display after line 415 (same for the first line in the display after line 417). Both of these rely on lemma A.4 and so I believe they should also pick up a term that involves $s^* \vee s' \log p / n$.  This obviously does not affect the result (it gets absorbed into the constant hidden by the $\preceq$ notation at the bottom of each display) but it did make it difficult to follow the argument.  Along these lines, is Lemma A.4 missing an absolute value sign on the right hand side of both equations in the displace below line 408?

* The results in Figure 5 suggest that it may be beneficial to continue increasing \lambda beyond the range of values considered in the figure.  In most rows, the power continues to increase as \lambda/\lambda_{univ} increases up to the final value displayed; additionally, the FDR remains within estimation error of being well-controlled.  That is, with 100 simulations and a true FDR of 0.1, one might expect a 95% confidence interval to be estimated_FDR +/- 1.96 * sqrt(0.1 * 0.9 / 100), which is ~0.06 so any FDR below 0.106 should be tolerable.  The authors should consider continuing to increase lambda to see if further power can be gained or if there is a point at which the regularization is too strong.

* In appendix G it's stated that the same clustering as 4.3 is used to reduce the dimension.  In F.1 its stated that the clustering preserves the spatial structure of the data.  Is that also the case for the genes in appendix G?  There is some clustering of genes by function in the genome, but overall not really so I'm not sure that it makes sense to include any spatial component here.  Furthermore, if the genes are clustered then why does Table 3 say n=1026 and p=24776, doesn't this contradict that the genes were clustered to 1000 clusters (p=1000)?  Finally, if the genes were clustered, then how are individual genes pulled out in Table 3?  Do all genes in a significant cluster get put in the table?





Typos:
* Line 13 "the geometry of ell_1" --> "the geometry of the ell_1"
* Line 29 "conditionally to" sounds off to me. Perhaps "conditioned on"?
* Lines 37-39 are difficult to parse grammatically
* Lines 54-55 "convergence to Gaussian"
* Lines 64-65: "standard normal test-statistic" --> either "a standard normal test-statistic" or "standard normal test-statistics"
* Line 65 "in large-sample regime" --> "in the large-sample regime"
* Figure 1 caption: "The empirical distribution of dCRT null-statistic" --> "The empirical distribution of the dCRT null-statistic"
* Line 71: "multiple sampling" --> "multiple samplings"
* Line 74: "hence inherently" --> "and hence inherently"
* Line 169: "where the formula for empirical" --> "where the formula for the empirical"
* Line 173-174: "We provide empirical benchmark of runtime" --> "We provide an empirical benchmark of the runtime"
* Line 176: "inline" --> "in line"
* Line 198: "using Benjamini-Yekutieli" --> "using the Benjamini-Yekutieli"
* Line 215: "implementation are" --> "implementation is"
* Line 253: "at default value" --> "at a default value"
* Lines 253-254: "using Benjamini-Hochberg procedure" --> "using the Benhamini-Hochberg procedure"
* Line 254: "Results in Figure 3" --> "The results in Figure 3"
* Line 310: "analyze two sample image" --> "analyze two sample images"
* Line 315: "is in emotion task" --> "is in the emotion task"
* Line 315: "failing to control FDR" --> "failing to control the FDR"
* Line 316: "at nominal level" --> "at the nominal level"
* Line 338: "We note that there exists" --> "We note that there exist"
* Line 404: "under logistic model" --> "under the logistic model"
* Line 405: "under logistic model" --> "under the logistic model"
* Line 413: "written in more general" --> "written in a more general"
* Line 415: "where we use triangle inequality" --> "where we use the triangle inequality"
* Line 416: "last inequality us due" --> "last inequality is due"
* Line 456: "from inference algorithm" --> "from an inference algorithm"
* Display below line 458: \hat{k}_{BY} should be \hat{k}_{BH}

**Limitations:**

I believe the authors have adequately addressed the limitations of their work, and I am not aware of any potential negative societal impact.

**Strengths And Weaknesses:**

The proposed approach is interesting and theoretically well-motivated.  That it gives (asymptotically) valid p-values is a compelling advantage over the knockoff framework that only allows for FDR control.  The asymptotics are also nice because then one can avoid needing to do any resampling.  One aspect that I found missing from the paper is a discussion of (the lack of) finite-sample guarantees.  The lack of finite sample guarantees is not a concern to me, but given that that's one of the compelling aspects of some of the related works (e.g., knockoffs) it would be good to at least discuss.

One minor weakness is that the clarity of the manuscript could be improved in some places.  It is quite dense (which is understandable due to space constraints).  While some parts of the paper have really nice intuitive explanations, other parts do not (e.g., equations 4 and 5).  Furthermore, the notation seems to be shifting throughout (see some of my comments in the "Questions" section) which can make it somewhat difficult to follow.

---

> ### Author Response · Authors · 2022-08-02
> **Answer to Reviewer np9G (part 1)**
>
> We thank the reviewer for a very detailed comments and spotting
> typos/incoherent notations in the manuscript.
>
> > One minor weakness is that the clarity of the manuscript could be improved in
> > some places. It is quite dense (which is understandable due to space
> > constraints). While some parts of the paper have really nice intuitive
> > explanations, other parts do not (e.g., equations 4 and 5). Furthermore, the
> > notation seems to be shifting throughout (see some of my comments in the
> > "Questions" section) which can make it somewhat difficult to follow.
>
> We agree with the comment of the reviewer. We have made changes on the revsions of the paper, and give some more details on the intuition of Eq(4) and (5), in particular the test statistics for dCRT is just a measure of Pearson correlation between regression residuals, scaled by a factor of $\sqrt{n}$ (elaboration put under Eq.(5)).
>
> We also fixed the incoherence in notations, especially in the proof in the Appendix following the reviewer's suggestions.
>
> > Figure 2 is interesting, but the asymptotics presented here -- and the really
> > interesting case -- is considering what happens as both n and p get large. A
> > very small simulation study to show that the asymptotic results hold even for
> > p >> n would, in my opinion, strengthen the work. Furthermore, in both the
> > brain processing application and the TCGA application the data is
> > preprocessed so that p ~ n. Is this because the p^3 scaling makes it
> > infeasible to run on the full dataset, or are there additional issues with $p \gg n$?
>
> Indeed, the reviewer makes a good remark on the computational reasoning: the
> runtime with $\mathcal{O}(p^3)$ makes it infeasible to run on the full dataset,
> with $p=200,000$ on the brain-imaging and $p=24766$ on genomics dataset.
>
> Following this argument, and to further answer the question of the reviewer on
> the statistical power, we have added in the Appendix (Section I) an extra simulation
> scenario with the same settings as in Section 4.1 and 4.1, but with fixed
> number of samples $n=400$ and varying dimension
> $p=(200,400,800,1600,3200,6400)$. The result shows that the when $p \gg n$, at
> the level of controlling FDR=0.1, the Average Power (on 100 simulations)
> decreasing to 0.0 when p reaches 1600. We believe this is due to the fact that
> in these scenarii, the sparsity estimator used to estimated the logistic
> regression, and for the decorrelation step cannot perform well due to very
> limited number of data points provided.
>
> More simply, on top of algorithmic considerations, it is obvious that the conditional inference problem only becomes harder when p increases. This is also why we has to resort to dimension reduction procedures in the experiments on real data.
>
>
> > There is no discussion of finite sample guarantees. e.g., Knockoffs control
> > the FDR for finite sample sizes, whereas the present method only has such
> > guarantees in the asymptotic regime. This is not a big deal, but it would be
> > good to mention this somewhere (maybe in the related works section).
> >
>
> Indeed, there is no finite sample guarantees with the proposed approach.
> We agree with the reviewer on these two remarks, and have added the lack of finite-sample guarantee on Remark 3.1.
>
> > In lines 171-174 the mention of a variable-screening step makes it sound like
> > a somewhat heuristic procedure, when, in fact, it just leverages that it is
> > trivial to compute p-values for j's where is zero.
> >
>
> > In Theorem 3.2, is $p$ overloaded as both the number of features in the
> > design matrix and also the number of tests? If separate they should not both
> > be labeled $p$. If they are the same, then isn't the theorem trivial by the
> > assumption of being fixed (eventually and it is essentially just standard
> > logistic regression)
> >
>
> We indeed assumed the number of tests is the same as the number of variables,
> which means they are both label $p$ purposefully. We will add a remark on this
> in the revised version. As the second part of the remark, we adjust the Theorem
> 3.2 to become a Corollary of Theorem 3.1, to reflect that it is a consequence.
>
> > Remark 4.2 feels important and should be emphasized a bit more in my opinion
> > -- the procedure used in the empirical tests is not proven to be valid by the
> > analysis, but it seems to work in practice.
>
> Indeed, this is correctly pointed out by the reviewer. We have also added a mentioning of this remark in the Discussion section.
>
> We remark that we are not able to prove independence/PRDS but BH still seems to
> work well; intuitively, the conditional test exhausts dependency with other
> variables $-j$ on each of the $j$, so the procedure may achieve
> quasi-dependence, but we were not able to rigorously
> prove this yet, and this might be a good future direction in terms of
> theoretical perspective of the procedure.

---

> > ### Comment · Reviewer_np9G · 2022-08-04
> > **Response to response**
> >
> > Thank you for the through response and the interesting paper.  I've increased my score to 7 based on the response and the revised manuscript, which fully addressed my primary comments.

---

> ### Author Response · Authors · 2022-08-02
> **Answer to Reviewer np9G (part 2)**
>
> > The notation in appendix A was a bit confusing to me. In particular, in
> > several places (e.g., the definition of $v^*$) a vector is defined which only
> > makes sense later under the assumption that $j=1$. For example $v^*$ is
> > defined as (1, $-w^{0, j}$, but then $v^{*T}\nabla\ell(\beta)$ only makes
> > sense if the 1 in lines up with $j$. This is obviously not a big problem and
> > doing something more "correct" might even be notationally clunky enough to
> > not be worth it, but at the very least a quick explanation of the abuse of
> > notation would be helpful.
>
> This is a good remark, and indeed we have made an abuse of the notation. As the
> reviewer pointed out, we used the fact that permutation invariant of the
> variable index in $\hat{v}$ or $v^*$ and correspondingly with
> $\nabla\ell(\beta)$ does not change the value of $v^T\nabla\ell(\beta)$, and we have added this reasoning to the Revised Appendix as Remark A.1.
>
> > I may be missing something, but is some term missing from the first line in
> > the display after line 415 (same for the first line in the display after line
> > 417). Both of these rely on lemma A.4 and so I believe they should also pick
> > up a term that involves $(s^* \vee s') \log(p) / n$. This obviously does not
> > affect the result (it gets absorbed into the constant hidden by the
> > $\precsim$ notation at the bottom of each display) but it did make it
> > difficult to follow the argument.
>
> The reviewer is correct: there is missing term $(s^* \vee s') \log(p) / n$
> (from Lemma A.4), we have updated this in the revised Appendix.
>
> > Along these lines, is Lemma A.4 missing an absolute value sign on the right
> > hand side of both equations in the displace below line 408?
>
> Indeed, the absolute value sign is missing on the LHS of the lemma -- we fixed
> this in the revised Appendix.
>
> > The results in Figure 5 suggest that it may be beneficial to continue
> > increasing \lambda beyond the range of values considered in the figure. In
> > most rows, the power continues to increase as \lambda/\lambda_{univ}
> > increases up to the final value displayed; additionally, the FDR remains
> > within estimation error of being well-controlled. That is, with 100
> > simulations and a true FDR of 0.1, one might expect a 95% confidence interval
> > to be estimated_FDR +/- 1.96 * sqrt(0.1 * 0.9 / 100), which is ~0.06 so any
> > FDR below 0.106 should be tolerable. The authors should consider continuing
> > to increase lambda to see if further power can be gained or if there is a
> > point at which the regularization is too strong.
>
> Following the reviewer's suggestion, we reran the experiment with wider value
> grid for the regularization parameter $\lambda$ (up to
> $log_{10}(\lambda/\lambda_{univ}) = 5.0$), but found no significant increase
> in terms of performance on avg. statistical power vs. FDR control (figure 5 updated). A slight
> caveat is that due to the time constraint and the large number of
> hyperparameter grids of this experiment, we only reran this experiment with 50
> simulations (on fixed p=400; 6 different n_samples; 17 different lambdas).
>
> Interestingly, this did not reveal new behavior, confirming that the main effect of lambda is a transition from a low-power to high-power situation, that saturates yet not exceeds the nominal error rate.
>
> > In appendix G it's stated that the same clustering as 4.3 is used to reduce
> > the dimension. In F.1 its stated that the clustering preserves the spatial
> > structure of the data. Is that also the case for the genes in appendix G?
> > There is some clustering of genes by function in the genome, but overall not
> > really so I'm not sure that it makes sense to include any spatial component
> > here.
>
> It is true that the way we wrote in the preprocessing step of the genomics data
> makes it confusing: it is only the same as in brain-imaging case as to reduce
> the effective dimension. While we still use clustering to reduce the dimension,
> we use different criterion to merge variables (genes) to clusters of variables,
> which is pairwise Linkage Disequilibrium, following [ADNRV19, Section 4] (with
> available public R library). We have added this elaboration to the revised
> edition of the manuscript.
>
> > Furthermore, if the genes are clustered then why does Table 3 say n=1026 and
> > p=24776, doesn't this contradict that the genes were clustered to 1000
> > clusters (p=1000)?
>
> This is a typo by our part, we fixed it in the revision of the manuscript.
>
> > Finally, if the genes were clustered, then how are individual genes pulled
> > out in Table 3? Do all genes in a significant cluster get put in the table?
>
> It is indeed the case here: we make inference on 1000 clusters, then all genes
>
>
> [ADNRV19]: Ambroise, C., Dehman, A., Neuvial, P., Rigaill, G., & Vialaneix,
> N. (2019). Adjacency-constrained hierarchical clustering of a band similarity
> matrix with application to genomics. Algorithms for Molecular Biology, 14(1),
> 1-14.

---

### Official Review · Reviewer_XeSh · 2022-07-11

**Rating:** 7
**Confidence:** 3
**Soundness:** 3 good
**Presentation:** 3 good
**Contribution:** 3 good

**Summary:**

The authors study the case of high-dimensional logistic regression when the number of features p is much greater than the number of sample n.  They propose the CRT-logit algorithm that combines a variable-distillation step and decorrelation step to keep the sparsity in l1-penalized logistic regression.  The authors provide theoretical analysis of their approach and show it is effectiveness on simulations and experiments on real-world brain-imaging and genomics data. The main contribution is the proposal of the CRT-logit method which is shown to be effective and inference cost not prohibitively high.

**Questions:**

1. Why do we need the noise term in Eqn. 12?
2. Can this method be generalized to work on a family class of models in high dimensions, e.g. any generalized linear model (GLM)?
3. What happens in the other case when n >> p?

**Limitations:**

Yes the authors have addressed the limitations of their work.

**Strengths And Weaknesses:**

Strengths
- The authors propose CRTlogit and provide thorough theoretical and experimental results
- Highly relevant problem since logistic regression is still one of the most commonly used methods as well as l1-penalty in machine and deep learning
- Valid and thorough theoretical results
- Show empirical validation of theoretical results
- Also show experiments on average inference runtime in addition to performance results
- Tests on real-world brain and genomics data.
- Very nice qualitative plots e.g. Figure 1 showing the improvement on the theoretical quantiles of the proposed CRT-Logit over dCRT.

Weaknesses
- Some of the derivations in Eqn, 8-11 can be moved to the appendix.
- Limited to only looking at logistic regression - can this extend to other models?
- The authors should emphasize their novelty and contribution as more than just an extension of CRT.

---

> ### Author Response · Authors · 2022-08-02
> **Answer to Reviewer XeSh**
>
> We thank the reviewer for his/her remarks. Followings are our answers to the
> questions.
>
> > Limited to only looking at logistic regression - can this extend to other
> > models?
>
> > Can this method be generalized to work on a family class of models in high
> > dimensions, e.g. any generalized linear model (GLM)?
>
> Indeed, the reviewer is correct that extension of this work to some families of
> generalized linear model proceeds naturally, as it is natively supported by the analysis in [NL17]. We have added a brief remark in the discussion area for this perspective.
>
> > The authors should emphasize their novelty and contribution as more than just
> > an extension of CRT.
>
> We agree with the reviewer. In fact, in the contribution paragraph in Section
> 1, we noted that we adapt the dCRT for the classification case (under logistic
> relationship), while improving on the prohibitive computational cost of the
> original CRT.
>
> > Why do we need the noise term in Eqn. 12?
>
> The noise term is used to increase the difficulty of the logistic regression
> problem -- we basically introduced an SNR parameter that dictates the
> signal-to-noise ratio of the simulated dataset. We also have pointed out in the
> remark below that the noise term can be understood as measurement error in data
> collection for realistic scenario. Technically, it results in the model being mis-specified, but the approach followed is robust to this mis-specification (see l.229-231).
>
> > What happens in the other case when n >> p?
>
> As Theorem 3.1 and 3.2 pointed out, when $n \gg p$, we will have the test
> statistics with empirical distribution almost perfectly similar to standard
> normal distribution; and so we no longer need the correction for the test-statistics.
>
> [NL17]: Ning, Y., & Liu, H. (2017). A general theory of hypothesis tests and confidence regions for sparse high dimensional models. The Annals of Statistics, 45(1), 158-195.

---

> > ### Comment · Reviewer_XeSh · 2022-08-08
> > **Response to response**
> >
> > Thank you to the authors for provided detailed answers to all of the reviewers and the revised manuscript. I have no further questions, and my concerns were addressed.

---

### Official Review · Reviewer_7y9r · 2022-07-11

**Rating:** 6
**Confidence:** 3
**Soundness:** 3 good
**Presentation:** 3 good
**Contribution:** 2 fair

**Summary:**

This paper aims to extend the growing literature on identifying relevant features in a machine learning model. The authors focus on the setting of classification in a high-dimensional setting where the number of relevant features is sparse (i.e., less than n^{1/2}), where n is the number of measurements and p is the number of features. To do so, they extend the algorithm of distilled conditional randomization test (dCRT), which itself is an extension of the conditional randomization test (CRT) to make CRT more computationally feasible.

A key step of the dCRT algorithm is to take residuals from two regressions (one regression to see if feature j lies in the linear span of the other features, and the second regression to see if the response variable lies in the linear space of the other features) and use that to compute the test statistic. In essence the key innovation of this paper is to extend the way the residuals are taken to fit a logistic model rather than a linear model. This estimator is called CRT-logic.

Given this new test statistic and an assumption of sparsity along with other regularity conditions, they prove pointwise gaussian approximation and asymptotic validity of the CRT-logit estimator. They then corroborate their results with simulations and real-world data.


**Questions:**

1.	What does the notation  y_i(X^T_{I, -j} \beta) mean? Is that an argument to y_i() or multipled by y_i? For example see (4)
2.	For the comparison with dCRT in Figure 1, how does dCRT perform when not using the original features, but a dictionary mapping of the original features (e.g. low-degree polynomial)? This might be a fairer comparison as it is quite obvious that just using the residuals from doing linear regression on the original features will lead to a poor for fit for a logistic function.
3.	Is the result in Theorem 3.1 assuming that the solve_scaled_lasso_cv step works perfectly? The result in Theorem 3.2 is for the expected FDR. Can that result be used as is in Theorem 3.1? I don’t see how.


**Strengths And Weaknesses:**

Strengths

-	The paper tackles an important problem of understanding feature relevance in a high-dimensional classification setting with sparse features.
-	The empirical results support their claim
-	The paper is relatively straightforward to parse through.


Weaknesses

-	From a theoretical standpoint, more work can be done in explaining the technical novelty in extending the method of dCRT to the setting of classification. Right now the revised estimator, which used the second derivative of the logistic function instead, has minimal intuition for how it is derived, and what makes this problem a technical challenge.
-	It is unclear which part of the estimator / results are to do with increasing computational efficiency vs. novel statistical results for the high-dimensional classification setting. This also makes it hard to tease out how to evaluate the estimator and the results.

---

> ### Author Response · Authors · 2022-08-02
> **Answer to Reviewer 7y9r**
>
> We thank the reviewer for his/her remarks. Followings are our answers to the
> questions.
>
> > From a theoretical standpoint, more work can be done in explaining the
> > technical novelty in extending the method of dCRT to the setting of
> > classification.
>
> We agree with the reviewer: we extend the paragraph below equation (5) to state that this equation (which is
> proposed as a test statistics for dCRT) is basically a calculation of
> correlation of the regression residuals, then scaled by a factor of
> $\sqrt{n}$. Hence, the correct distribution of the statistic defined by the formula depends heavily on
> how well the residuals are estimated.  This becomes non-trivial when the
> relationship between labels and features become non-linear, e.g. in
> classification setting with logistic relationship.
>
> > Right now the revised estimator, which used the second derivative of the
> > logistic function instead, has minimal intuition for how it is derived, and
> > what makes this problem a technical challenge.
>
> We believe that the intuition of the novelty of the method have been elaborated
> in the beginning of Section 3: observing that the Fisher score (the gradient of the negative log-likelihood wrt to parameters, i.e. the natural decision statistic for the problem) is biased in the
> $n < p$ setting, we correct that score.
>
> > It is unclear which part of the estimator / results are to do with increasing
> > computational efficiency vs. novel statistical results for the
> > high-dimensional classification setting. This also makes it hard to tease out
> > how to evaluate the estimator and the results.
>
> Regrding the statistical efficiency of the decorrelating statistics, we argued
> in Eq.(7) that the second term on the right-hand side is not negligible in the
> $n < p$ case, and therefore we try to cancel the effect of this term by
> subtracting an empirical esimation of it.
>
> Regarding computational efficiency, we do not claim that our method is faster than the original dCRT. In fact, the iteration
> complexity of the two methods should be the same, as stated in the Appendix.
> The benchmarked runtime of Table 1 in the main text also supports this
> analysis. We also make a remark on the computational cost of CRT-logit in
> the Discussion, and note that it is one of the main limitations of
> CRT-logit. This can be partially addressed by running the computation
>  for each variable in parallel.
>
> > What does the notation $y_i (X^T_{I, -j} \beta)$ mean? Is that an argument to
> > $y_i()$ or multipled by $y_i$? For example see (4)
>
> We agree that this kind of notation might be confusing: what we meant is the
> latter case. To avoid this minor confusion we swap the case, and use $(X^T_{I, -j}
> \beta) y_i $ instead in the revision of the paper.
>
> > For the comparison with dCRT in Figure 1, how does dCRT perform when not
> > using the original features, but a dictionary mapping of the original
> > features (e.g. low-degree polynomial)?
>
> We agree with the reviewer that performing adequate dimension reduction is essential in problems with $n \ll p$.
> This is actually what we do in the brain imaging and geentic experiments, where we consider clustering-based dimension reduction.
> The only caveat is that one should be aware on what objects precisely the null hypothesis is rejected: not original features, but combinations thereof (dictionary elements, clusters etc.). We make the motivation for such reduction more explicit in the revised version (appendix I), by showing that the approach becomes powerless whenever $n \ll p$.
>
> Could the reviewer elaborate on the dictionary mapping with low-degree polynomial? As far as we are aware, when we include a fitted model with low-degree polynomial, the number of parameters will actually increase, making the dimension issue actually worse ?
>
> > This might be a fairer comparison as it is quite obvious that just using the
> > residuals from doing linear regression on the original features will lead to
> > a poor for fit for a logistic function.
>
> We agree that it can be obvious, but we think that this is the whole motivation
> of the manuscript: adapting the obviously biased estimation of the dCRT to the
> case of non-linear relationship.
>
> >  Is the result in Theorem 3.1 assuming that the solve_scaled_lasso_cv step works perfectly?
>
> Indeed this is the case, and we reflect it in Assumption 3.1-(A2), which is used for Lemma A.3 (that states the bound for estimation error of the scaled lasso) in the Supplementary Material.
>
> > The result in Theorem 3.2 is for the expected FDR. Can that result be used as
> > is in Theorem 3.1? I don’t see how.
>
> Perhaps there is a typo in the reviewer comment, as you would mean expected FDP
> (since the FDR=E[FDP]) in asymptotic case? In fact, in our proof for Theorem
> 3.2, we need to use the asymptotic results in Theorem 3.1. We have added a discussion (as pointed out by reviewer np9G) that we have not found a finite-sample guarantee for the theoretical analysis of the decorrelated test-score, and will leave it for future work.

---

### Official Review · Reviewer_VbNR · 2022-07-17

**Rating:** 7
**Confidence:** 2
**Soundness:** 3 good
**Presentation:** 3 good
**Contribution:** 3 good

**Summary:**

The authors study the problem of doing inference on a logistic regression model with an L1 penalty in high dimensions, where the number of features in the dataset is at least as large as the size of the dataset. For this problem, they develop a variant (that they call CRT-logit) of the distilled conditional randomization test (dCRT) with higher power than the latter. Their innovation is in introducing a decorrelation step that brings the null distribution of the test statistic closer its assumed distribution - a standard normal. An asymptotic analysis of the performance of CRT-logit is given.

**Questions:**

CRT-logit is proposed for problems in which $p>n$, but the authors give an asymptotic ($n\rightarrow\infty$) analysis of its performance in Theorem 3.1. At a high level, I am confused by this meta structure in the paper. Do the authors have a sense for how quickly (in $n$) the p-values supplied by the algorithm become ‘good’ in some sense?

What is the utility of adding the extra assumption of sub-exponential tail behavior only in the appendix (I am comparing Assumption A.1, (3) to Assumption 3.1, (3))?

There is a typo in line 93. $x$ should be $x_n$.

The ‘description paragraph’ in section 4.3 could be made clearer. In particular, I suggest the authors simply state that they are working with fMRI data and that the goal of the analysis is to identify voxels with task-related levels of activity. As written, the type of data and the goal of the analysis are unclear.

**Limitations:**

Yes.

**Strengths And Weaknesses:**

I think this is a very nice paper. It focuses on an important and ubiquitous inference problem, and is generally of a high quality, largely due to its clarity and thoroughness. The central technical innovation, namely, the decorrelation procedure discussed in Eqs. 8-11, is well-motivated both analytically and in Figure 1. The new algorithm that is introduced and tested, CRT-logit, will be of interest to the broader machine learning community. Its relationship to existing algorithms is discussed; a number of experiments comparing it with existing algorithms are also presented.

---

> ### Author Response · Authors · 2022-08-02
> **Answer to reviewer VbNR**
>
> We thank the reviewer for his/her remarks. Followings are our answers to the
> questions.
>
> > CRT-logit is proposed for problems in which $n > p$, but the authors give an
> > asymptotic ($n \to \infty$) analysis of its performance in Theorem 3.1. At a
> > high level, I am confused by this meta structure in the paper. Do the authors
> > have a sense for how quickly (in $n$) the p-values supplied by the algorithm
> > become ‘good’ in some sense?
> >
>
> Indeed, due to the lack of space, we have not provided a more detailed comment
> on the speed of convergence in the main text. We put it in the proof of Theorem
> 3.1, which shows that the p-values "becomes good" at the rate
> $\mathcal{O}(1 / \sqrt{n})$.
>
> We agree that it is better to put this in the main text, and have updated it
> accordingly under Remark 3.1.
>
> > What is the utility of adding the extra assumption of sub-exponential tail
> > behavior only in the appendix (I am comparing Assumption A.1, (3) to
> > Assumption 3.1, (3))?
> >
>
> It is true that it might get confusing to compare the assumption A.1 and 3.1 in the main text: the sub-exponential tail behavior is needed for our proof of Theorem 3.1. We noted in Remark A.1 (the beginning of SupMat file) that this is purely for theoretical analysis only, and will not affect experimental results section. This is a quite standard requirement for this type of analysis.
>
> We have merged these two assumptions in the main text with the revised version, in particular, we added the sub-exponential to Assumption 3.1 in the main text.
>
> > The ‘description paragraph’ in section 4.3 could be made clearer. In
> > particular, I suggest the authors simply state that they are working with
> > fMRI data and that the goal of the analysis is to identify voxels with
> > task-related levels of activity. As written, the type of data and the goal of
> > the analysis are unclear.
>
> We agree with the reviewer comment, and added this sentence in the Description paragraph of Section
> 4.3 following the suggestion.

---

> > ### Comment · Reviewer_VbNR · 2022-08-08
> > **Re: author response**
> >
> > I thank the authors for their responses and for editing the paper. I have no further questions at present.

---

### Author Response · Authors · 2022-08-02
**General answers to all reviewers**

We thank the reviewers for their efforts to provide very detailed comments and
suggestions. We take into account the comments on typos of the manuscript, and
we have taken these into account with the revised version. A slight caveat is
that at the moment, due to the strict 9-pages constraint of the revised-edition, we cannot add all of the suggestions/remarks inside the
revised version of the main text yet, but we promise to do that in the future
iteration of the manuscript.

We have provided dedicated answers to reviewers accordingly, following each of
their reviews.

---

### Meta-Review · Area_Chair_X5iM · 2022-08-27

**Recommendation:** Accept
**Confidence:** Certain

**Metareview:**

The decision is to accept this paper.

The paper presents a method for producing asymptotically valid p-values when testing the null hypothesis of conditional randomization tests in sparse logistic regression. The method builds on a previous distillation method that examines correlations between residuals for the label y and the focal covariate x_j when they are projected onto the remaining covariates. The method corrects a bias that arises in this distillation method due to the non-linearity in penalized logistic regression. The authors prove the asymptotic validity of the resulting p-values and study the power and FDR of the procedure.

The reviewers agreed that this is a strong method and a clearly written paper. The authors answered all major questions from the reviewers and made changes in response to reviewer feedback.

**Award:**

No

---

### Decision · Program_Chairs · 2022-09-14

Accept